# Autonomous Reinforcement Learning via Subgoal Curricula

**Archit Sharma**[†‡], **Abhishek Gupta**[#‡], **Sergey Levine**[#‡], **Karol Hausman**[‡†], **Chelsea Finn**[†‡]
[†] Stanford University, [‡] Google Brain, [#] UC Berkeley
{architsh,cbfinn}@stanford.edu
{abhishekunique,slevine,karolhausman}@google.com

## Abstract

Reinforcement learning (RL) promises to enable autonomous acquisition of complex behaviors for diverse agents. However, the success of current reinforcement learning algorithms is predicated on an often under-emphasised requirement – each trial needs to start from a fixed initial state distribution. Unfortunately, resetting the environment to its initial state after each trial requires substantial amount of human supervision and extensive instrumentation of the environment which defeats the goal of autonomous acquisition of complex behaviors. In this work, we propose Value-accelerated Persistent Reinforcement Learning (VaPRL), which generates a curriculum of initial states such that the agent can bootstrap on the success of easier tasks to efficiently learn harder tasks. The agent also learns to reach the initial states proposed by the curriculum, minimizing the reliance on human interventions into the learning. We observe that VaPRL reduces the interventions required by three orders of magnitude compared to episodic RL while outperforming prior state-of-the art methods for reset-free RL both in terms of sample efficiency and asymptotic performance on a variety of simulated robotics problems[1].

## 1 Introduction

Reinforcement learning (RL) offers an appealing opportunity to enable autonomous acquisition of complex behaviors for interactive agents. Despite recent RL successes on robots [26, 34, 25, 28, 35, 22, 32, 23, 14], several challenges exist that inhibit wider adoption of reinforcement learning for robotics [48]. One of the major challenges to the autonomy of current reinforcement learning algorithms, particularly in robotics, is the assumption that each trial starts from an initial state drawn from a specific state distribution in the environment. Conventionally, reinforcement learning algorithms assume the ability to arbitrarily sample and reset to states drawn from this distribution, making such algorithms impractical for most real-world setups.

Many prior examples of reinforcement learning on real robots have relied on extensive instrumentation of the robotic setup and human supervision to enable environment resets to this initial state distribution. This is accomplished through a human providing the environment reset themselves throughout the training [8, 12, 4], scripted behaviors for the robot to reset the environment [28, 39], an additional robot executing scripted behavior to reset the environment [32], or engineered mechanical contraptions [46, 23]. The additional instrumentation of the environment and creating scripted behaviors are both time-intensive and often require additional resources such as sensors or even robots. The scripted reset behaviors are narrow in application, often designed for just a single task or environment, and their brittleness mandates human oversight of the learning process. Eliminating or minimizing the algorithmic reliance on the reset mechanisms can enable more autonomous learning, and in turn it will allow agents to scale to broader and harder set of tasks.

---

[1]Code and supplementary videos are available at `https://sites.google.com/view/vaprl/home`

35th Conference on Neural Information Processing Systems (NeurIPS 2021).

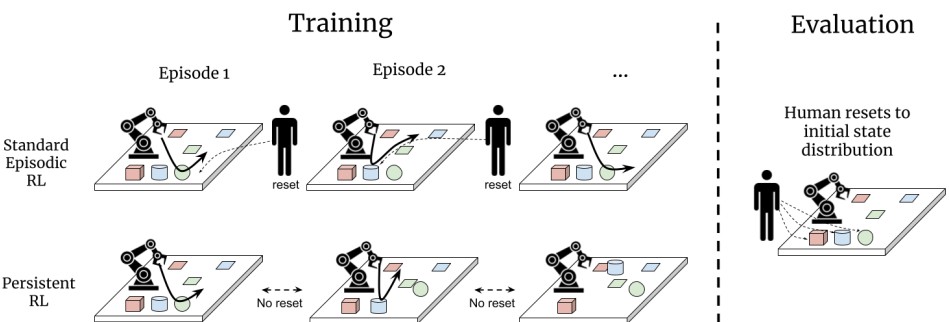

Figure 1: Comparison of the persistent RL setting with the episodic RL setting. Interventions (human or otherwise orchestrated) reset the environment to the initial state distribution after every episode in episodic RL, while the state of the environment persists through the training in persistent RL. The learned policy is tested starting from the initial state distribution for both the settings.

To address these challenges, some recent works have developed reinforcement learning algorithms that can effectively learn with minimal resets to the initial distribution [19, 6, 48, 43, 14]. We provide a formal problem definition that encapsulates and sheds light on the general setting addressed by these prior methods, which we refer to as the *persistent reinforcement learning* in this work. In this problem setting, we disentangle the training and the test time settings such that the test-time objective matches that of the conventional RL setting but the train-time setting restricts access to the initial state distribution by giving a *low frequency* periodic reset. In this setting, the agent must persistently learn and interact with the environment with minimal human interventions, as shown in Figure 1. Conventional episodic RL algorithms often fail to solve the task entirely in this setting, as shown by Zhu et al. [48] and Figure 2. This is because these methods rely on the ability to sample the initial state distribution arbitrarily. One solution to this problem is to additionally learn a reset policy that recovers the initial state distribution [19, 6] allowing the agent to repeatedly alternate between practicing the task and practicing the reverse. Unfortunately, not only can solving the task directly from the initial state distribution be hard from an exploration standpoint, but (attempting to) return to the initial state repeatedly can be sample inefficient. In this paper, we propose to instead have the agent reset itself to and attempt the task from different initial states along the path to the goal state. In particular, the agent can learn to solve the task from easier starting states that are closer to the goal and bootstrap on these to solve the task from harder states farther away from the goal.

The main contribution of this work is *Value-accelerated Persistent Reinforcement Learning* (VaPRL), a goal-conditioned RL method that creates an adaptive curriculum of starting states for the agent to efficiently improve test-time performance while substantially reducing the reliance on extrinsic reset mechanisms. Additionally, we provide a formal description of the persistent RL problem setting to conceptualize our work and prior methods. We benchmark VaPRL on several robotic control tasks in the persistent RL setting against state-of-the-art methods, which either simulate the initial state distribution by learning a reset controller, or incrementally grow the state-space from which the given task can be solved. Our experiments indicate that using a tailored curriculum generated by VaPRL can be up to 30% more sample-efficient in acquiring task behaviors

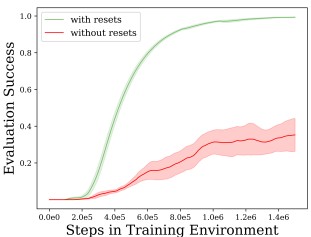

Figure 2: The performance of episodic RL algorithms substantially deteriorates when environment resets are not available.

compared to these prior methods. For the most challenging dexterous manipulation problem, VaPRL provides a 2.5× gain in performance compared to the next best performing method.

## 2 Related Work

**Robot learning.** Prior works using reinforcement learning have relied on manually design controllers or human supervision to enable episodic environmental resets, as is required by the current algorithms. This can be through human orchestrated resets [8, 13, 12, 4, 16], which requires high frequency human intervention in robot training. In some cases, it is possible to execute a scripted behavior to reset the environment [28, 32, 47, 39, 46, 1]. However, programming such behaviors is time-intensive for the practitioner, and robot training still requires human oversight as the scripted behaviors are often brittle. Some prior works have designed the environment [35, 7, 22] to bypass the need for having a reset mechanism. This is not generally applicable and can require extensive environment

design. Some recent works leverage multi-task RL to bypass the need for extrinsic reset mechanisms [15, 14]. Typically, a task-graph uses the current state to decides the next task for the agent, such that only minimal intervention is required during training. However, these task-graphs are specific to a problem and require additional engineering to appropriately decide the next task.

**Reset-free reinforcement learning.** Constraining the access to these orchestrated reset mechanisms severely impedes policy learning when using current RL algorithms [48]. Recent works have proposed several algorithms to reduce reliance on extrinsic reset mechanisms by learning a reset controller to retrieve the initial state distribution [19, 6], by learning a perturbation controller [48], or by learning reset skills in adversarial games [43]. These works implicitly define a target state distribution for a reset controller: Han et al. [19], Eysenbach et al. [6] target a fixed initial state distribution; Zhu et al. [47] target a uniform distribution over the state space as a consequence of novelty-seeking behavior of the reset controller; and Xu et al. [43] target an adversarial form of initial state distribution to produce a more robust policy. In contrast, our proposed algorithm VaPRL generates a curriculum of starting states tailored to the task and agent's performance. Our experiments demonstrate that VaPRL outperforms these prior methods in both sample efficiency and absolute performance. Other recent work like [29] has considered combining model-based RL with unsupervised skill discovery to solve reset-free learning problems, but largely focus on avoiding sink states rather than attempting tasks repeatedly with a curriculum like VaPRL.

**Curriculum generation for reinforcement learning.** Curriculum generation is a crucial aspect of sample-efficient learning in VaPRL. Prior works have shown that using a curriculum can enable faster learning and improve performance [9, 10, 40, 30, 33, 27]. Task-tailored curriculum can simplify the exploration as it is easier to solve the task from certain states [21, 9] enabling faster progress on the downstream task. In addition to proposing a novel method for curriculum generation, we design it for the persistent RL setting without requiring the ability to reset the environment to arbitrary states as assumed by prior work.

**Persistent vs. lifelong reinforcement learning.** Prior reinforcement learning algorithms that reduce the need for oracle resets have relied on the problem setting of lifelong or continual reinforcement learning [41, 24], when the objective in practice is to learn episodic behaviors. Both the persistent RL and the lifelong learning frameworks do transcend the episodic setting for training, promoting more autonomy in reinforcement learning. However, persistent reinforcement learning distinguishes between the training and evaluation objectives, where the evaluation objective matches that of the episodic reinforcement learning. While the assumptions of episodic reinforcement learning are hard to realize for real-world training, real-world deployment of policies is often *episodic*. This is commonly true for robotics, where the assigned tasks are expected to be repetitive but it is hard to orchestrate resets in the training environment. This makes persistent reinforcement learning a suitable framework for modelling robotic learning tasks.

## 3 Persistent Reinforcement Learning

In this section, we formalize the persistent reinforcement learning as an optimization problem. The key insight is to separate the evaluation and training objectives such that the evaluation objective measures the performance of the desired behavior while the training objective enables us to acquire those behaviors, while recognizing that frequent invocation of a reset mechanism is untenable. We first provide a general formulation, and then adapt persistent RL to the goal-conditioned setting.

**Definition.** Consider a Markov decision process (MDP) $\mathcal{M}_E \equiv (\mathcal{S}, \mathcal{A}, p, r, \rho, \gamma, H_E)$ [37]. Here, $\mathcal{S}$ denotes the state space, $\mathcal{A}$ denotes the action space, $p : \mathcal{S} \times \mathcal{A} \times \mathcal{S} \mapsto \mathbb{R}_{\geq 0}$ denotes the transition dynamics, $r : \mathcal{S} \times \mathcal{A} \mapsto \mathbb{R}$ denotes the reward function, $\rho : \mathcal{S} \mapsto \mathbb{R}_{\geq 0}$ denotes the initial state distribution, $\gamma \in [0, 1]$ denotes the discount factor, and $H_E$ denotes the episode horizon. Our objective is to learn a policy $\pi$ that maximizes $J_E(\pi) = \mathbb{E}[\sum_{t=1}^{H_E} \gamma^t r(s_t, a_t)]$, where $s_0 \sim \rho(\cdot)$, $a_t \sim \pi(\cdot \mid s_t)$ and $s_{t+1} \sim p(\cdot \mid s_t, a_t)$, the episodic expected sum of discounted rewards.

However, generating samples from the initial state distribution $\rho$ invokes a reset mechanism, which is hard to realize in the real world. We want to construct a MDP $\mathcal{M}_T$ corresponding to our training environment which reduces invocations of the reset mechanism. To reduce such interventions, we consider a training environment $\mathcal{M}_T \equiv (\mathcal{S}, \mathcal{A}, p, \tilde{r}_t, \tilde{\rho}, \gamma, H_T)$ with episode horizon $H_T \gg H_E$. Näively optimizing $r$ can substantially deteriorate the performance of episodic RL algorithms, as shown in Figure 2 where we compare the evaluation performance with $H_E = 200$ when training in environments with $H_T = 200$ (with resets) versus $H_T = 200,000$ (without resets). Therefore,

it becomes beneficial to consider a surrogate reward function $\tilde{r}_t$ rather than just optimizing for $r$ naively. As a motivating example, consider a forward-backward controller which alternates between solving the task corresponding to $r$ and recovering the initial state distribution $\rho$. The surrogate reward function corresponding for this approach can be written as:

$$\tilde{r}_t(s_t, a_t) = \begin{cases} r(s_t, a_t) & t = [1, H_E], [2H_E + 1, 3H_E], \dots \\ r_\rho(s_t, a_t) & t = [H_E + 1, 2H_E], \dots \end{cases} \tag{1}$$

Here, $\tilde{r}$ alternates between the task-reward $r$ for $H_E$ steps and $r_\rho$ (which encourages initial state distribution recovery) for $H_E$ steps[2], also illustrated in Figure 3 (*right*). This surrogate reward function allows the agent to repeatedly practice the task, thus using the autonomous interaction more judiciously as compared to the naïve approach. Note, this $\tilde{r}$ loosely recovers the objectives used in some prior works [19, 6]. For a general time-dependent surrogate reward function $\tilde{r}_t$, we define the training objective as

$$J_T(\pi) = \mathbb{E}_{s_0 \sim \tilde{\rho}, a_t \sim \pi(\cdot|s_t), s_{t+1} \sim p(\cdot|s_t, a_t)} \Big[ \sum_{t=1}^{H_T} \gamma^t \tilde{r}_t(s_t, a_t) \Big] \tag{2}$$

where $\tilde{\rho}$ is the initial state distribution at training time (which does not need to match the evaluation-time initial state distribution $\rho$). The persistent RL optimization objective is to maximize $J_T(\pi)$ efficiently under the constraint that $J_E(\arg\max_\pi J_T(\pi)) = \max_\pi J_E(\pi)$. Intuitively, the objective encourages construction of a training environment that can recover the optimal policy for the evaluation environment. The primary design choice is $\tilde{r}_t$, which as shown above leads to different algorithms. Another design choice is $\tilde{\rho}$, which may or may not match $\rho$. Importantly, we do not assume $\tilde{\rho}$ is any easier to sample compared to $\rho$.

Finally, we note that the formulation discussed here is suitable only for reversible environments. Reversible environments guarantee that the agent can continue to make progress on the task and not get "stuck" (for example, if the object goes out of reach of the robot's arm). A large class of practical tasks can be considered reversible (door opening, cloth folding, and so on) or the environment can be constructed to enforce reversibility (add bounding walls so the object does not go out of reach). A formal definition for reversible environments is provided in Appendix A. In this work, we will restrict ourselves to reversible environments, and defer a full discussion of persistent RL for environments with irreversible states to future work.

**Goal-conditioned persistent reinforcement learning.** We adapt the general formulation above to a goal-conditioned [20, 38] instantiation of persistent RL. Consider a goal-conditioned MDP $\mathcal{M}_E \equiv (\mathcal{S}, \mathcal{A}, \mathcal{G}, p, r, \rho, \gamma, H_E)$, where $\mathcal{G} \subseteq \mathcal{S}$ denotes the goal space. For a goal distribution $p_g : \mathcal{G} \mapsto \mathbb{R}_{\geq 0}$, the evaluation objective is $J_E(\pi) = \mathbb{E}_{g \sim p_g(\cdot)} \mathbb{E}_{\pi(\cdot|s,g)} [\sum_{t=1}^{H_E} \gamma^t r(s, g)]$ for $\pi : \mathcal{S} \times \mathcal{A} \times \mathcal{G} \mapsto \mathbb{R}_{\geq 0}$. The training objective is then stated as:

$$J_T(\pi) = \mathbb{E}_{s_0 \sim \tilde{\rho}, a_t \sim \pi(\cdot|s_t, G(s_t, p_g)), s_{t+1} \sim p(\cdot|s_t, a_t)} \Big[ \sum_{t=1}^{H_T} \gamma^t r(s_t, G(s_t, p_g)) \Big], \tag{3}$$

where we assume that $\tilde{r} = r$ remains as a goal reaching objective, but where algorithms instead use a goal generator $G$ to generate a curriculum of goals to practice throughout training[3]. The intuition is that, since $H_T \gg H_E$, the algorithm can repeatedly practice reaching various task-goals. However, the objective is to learn a policy that can reach task-goals from $p_g$ in the test environment, i.e., starting from the initial state distribution $\rho$. This implies the goal generator $G$ should expand the goal space beyond the task-goals to improve the policy $\pi$ for the test environment. For example, the goal generator could alternate task-goals ($g \sim p_g$) and the initial state distribution ($s \sim \rho$), which again loosely recovers prior works [19, 6]. This instantiation transforms the problem of finding the right reward function $\tilde{r}$ to the right curriculum of goals using $G$.

## 4 Value-Accelerated Persistent Reinforcement Learning

To address the goal-conditioned persistent RL problem, we now describe our proposed algorithm, VaPRL. The key idea in VaPRL is that the agent does not need to return to the initial state distribution between every attempt at the task, and can instead choose to practice from states that facilitate efficient learning. Section 4.1 discusses how to generate this curriculum of initial states. Using

---

[2]We assume that the state includes information indicating the reward function being optimized so that agent can take appropriate actions, for example, one-hot task indicators as is common in multi-task RL.

[3]The goal generator may use additional memory which is not explicitly represented here.

VaPRL Training                    Forward–Backward RL

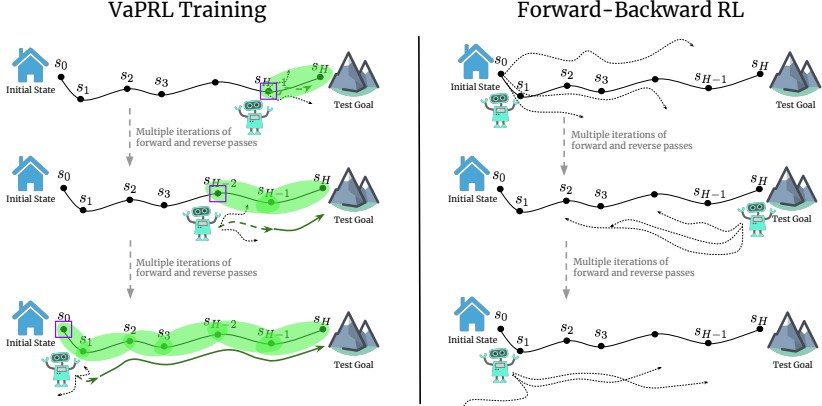

Figure 3: An overview of the VaPRL algorithm *(left)* compared to forward-backward RL *(right)*. For VaPRL, the value function gives us a set of states from where the agent can solve the task with some confidence (shaded in green), and the VaPRL chooses the state closest to the initial state distribution among them (purple square). In each iteration, the agent can bootstrap on the knowledge of solving the task from a future state (bold green) which simplifies the exploration from its current state (broken green line). As the performance of the agent improves, the states commanded by VaPRL move closer to the initial state distribution. This is in contrast to the forward-backward controller that alternates between the test-goals and the initial state distribution.

goal-conditioned RL within VaPRL allows us to use the same policy to solve the task and reach the initial states suggested by the curriculum, in contrast to prior work that learns a separate reset and task policy. Section 4.2 describes how careful goal relabeling can be leveraged to efficiently learn this unified goal-reaching policy. We also discuss how VaPRL can effectively use prior data, which often becomes crucial for efficiently solving hard sparse-reward tasks.

## 4.1 Generating a Curriculum Using the Value Function

Consider the problem of reaching a goal $g \sim p_g$ in the MDP $\mathcal{M}_E$. Learning how to reach the goal $g$ is easier starting from a state $s \in \mathcal{S}$ that is close to $g$, especially when the rewards are sparse. Knowing how to reach the goal $g$ from a state $s$ in turn makes it easier to reach the goal from states in the neighborhood of $s$, enabling us to incrementally move farther away from the goal $g$. Bootstrapping on the success of an easier problem to solve a harder problems motivates the use of curriculum in reinforcement learning, also illustrated in Figure 3.

Following the intuition above, we aim to define an increasingly-difficult curriculum such that the policy is eventually able to reach the goal $g$ starting from the initial state distribution $\rho$. Our simple scheme is to sample a task goal $g \sim p_g$, run the policy $\pi$ with a subgoal $C(g)$ as input, and then run the policy with the task goal $g$ as input. The main question now becomes: given a goal $g$, how do we select the subgoal $C(g)$ to attempt the goal $g$ from? We propose to set up $C(g)$ as follows:

$$C(g) = \arg\min_{s} \mathcal{X}_\rho(s) \quad \text{s.t.} \quad V^\pi(s,g) \geq \epsilon, \tag{4}$$

where $\mathcal{X}_\rho$ is a user-specified distance function between the state $s$ and the initial state distribution $\rho$, $V^\pi(s,g) = \mathbb{E}[\sum_{t=1}^{H_E} \gamma^t r(s,g) \mid s_1 = s]$ denotes the value function of the policy $\pi$ reaching the goal $g$ from the state $s$, and $\epsilon \in \mathbb{R}$ is some fixed threshold. Here, the value function represents the ability of the policy to reach the goal $g$ from the state $s$. To see that, consider the case when discount factor $\gamma = 1$ and $r(s,g) = 1$ when $s \approx g$ and 0 otherwise, the value function $V^\pi$ exactly represents the probability of reaching a goal $g$ from state $s$ when following the policy $\pi$. The intuition carries over to $\gamma \in [0,1)$ too, where the environment can go into a terminal state with probability $1 - \gamma$ at every transition. For general goal-reaching reward functions, a state $s$ with a higher value under $V^\pi(s,g)$ would still represent greater ability to reach the goal $g$ for the policy $\pi$.

Revisiting Equation 4 with this understanding of the value function, the objective $C(g)$ chooses the state closest to the initial state distribution for which the value function $V^\pi(s,g)$ crosses the threshold $\epsilon$. This encourages the curriculum to be closer to the goal state in the early stages of the training as the policy would be less effective at reaching the goal. As the policy improves, a larger number of states satisfy the constraint and the curriculum progressively moves closer to the initial state distribution. Eventually, the curriculum converges to the initial state distribution leading to a policy $\pi$ that would optimize the evaluation objective in the MDP $\mathcal{M}_E$. Following this intuition, we can write the goal

generator $G(s_t, p_g)$ as:

$$G(s_t, p_g) = g \quad \text{s.t.} \begin{cases} g_{task} \sim p_g, \ g \leftarrow C(g_{task}) & \text{if switch}(s_t, g) = \texttt{subgoal} \\ g \leftarrow g_{task} & \text{elif switch}(s_t, g) = \texttt{task goal} \end{cases} \quad (5)$$

where the switch$(s_t, g_{cur})$ is true if the $g_{cur}$ has been reached or $g_{cur}$ has been in place for $H_E$ steps. For every new goal $g_{task} \sim p_g$, we first attempt to reach the curriculum subgoal $C(g_{task})$ (that is switch$(s_t, g)$ = $\texttt{subgoal}$), and then we attempt to reach goal $g_{task}$ (that is switch$(s_t, g)$ = $\texttt{task goal}$). This cycles repeats until the environment resets after $H_T$ steps.

**Computing the Curriculum Generator $C(g)$.** Equation 4 involves a minimization over the state space $\mathcal{S}$, which is intractable in general. While it is possible to come up with general solutions by constructing a generative model $p(s)$ and taking a minimum over the samples generated by it, we opt for a simpler solution: we use the data collected by the policy $\pi$ during the training and minimize $C(g)$ over a randomly sampled subset of it by enumeration. If an offline dataset or demonstrations are available, we can also minimize $C(g)$ on this data exclusively. The constrained minimization can similarly be approximated by considering the subset of the data which satisfies the constraint $V^\pi(s, g) \geq \epsilon$ and choosing the state from this subset which minimizes $\mathcal{X}_\rho$. If no state satisfies the constraint, $C(g)$ returns the state with the maximum $V^\pi(s, g)$.

**Measuring the Initial State Distribution Distance.** An important component of curriculum generation is choosing the distance function $\mathcal{X}_\rho(s)$, which should reflect the distance to the state from the initial state distribution under the environment dynamics. To factor in the dynamics, we can use the learned goal-conditioned value function as measure of shortest distance between the state and the goal [20, 36]. In particular, we use $\mathcal{X}_\rho(s) = -\mathbb{E}_{s_0 \sim \rho} V^\pi(s, s_0)$. While this choice is convenient as we are already estimating $V^\pi(s, g)$, there is an even simpler choice for $\mathcal{X}_\rho(s)$ when offline demonstration data is available. Assuming that the trajectories in the provided dataset start from the initial state distribution $\rho$, we can use the timestep index of the state as the distance from the initial state distribution, that is $\mathcal{X}_\rho(s) = \arg_t(s, \mathcal{D})$ where $\mathcal{D}$ denotes the offline demonstration set. The *step index distance function* defined here encodes the intuition that states which require more steps by the policy are farther away. The function naturally accommodates for the dynamics of the environment. Finally, since we are minimizing $\mathcal{X}_\rho(s)$ in Equation 4, if there are multiple trajectories to the same state or suboptimal loops within a single trajectory, we use the shortest distance to that state.

### 4.2 Relabeling Goals

Not only does our policy need to learn how to reach $g \sim p_g$, but it also needs to learn how to reach all the goals generated by $C(g)$ over the course of training, causing the effective goal-space to grow substantially. However, there is a lot of shared structure in reaching goals, especially those generated by the curriculum. The knowledge of how to reach a goal $g_1$ also conveys meaningful information about how to reach a goal $g_2$. This structure can be leveraged by using techniques from goal relabeling [2]. In particular, we relabel every trajectory collected in the training environment with $N$ goals sampled randomly from the set of goals that may be a part of the curriculum. If we do not have any prior data, we randomly sample

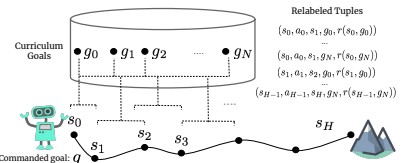

Figure 4: An illustration of goal relabeling in VaPRL. Every transition in a trajectory is relabeled with a randomly sampled subset of curriculum goals, yielding a large set of relabeled tuples that are added to the replay buffer. This ensures efficient data reuse.

the replay buffer for relabeling goals. If we are given some prior data $\mathcal{D}$, this reduces sampling to $g \sim \mathcal{D} \cup \{g' \sim p_g\}$ for relabeling.

There is a subtle difference between hindsight experience replay (HER) and the goal relabeling strategy we employ. While HER chooses future states from within an episode as goals for relabeling, we exclusively choose states that may be used as goal states in the curriculum, which may not occur in the collected trajectory at all. Since our policy will only be tasked with reaching goals generated by the goal generator $G$, it is advantageous to extract signal specifically for these goals. To summarize, while the goal-space has grown, goal relabeling enables us to generate data for the algorithm commensurately to improve sample-efficiency.

**Algorithm Summary.** The outline for VaPRL is given in Algorithm 1. At a high level, VaPRL takes a set of demonstrations as input and adds it to the replay buffer $\mathcal{R}$. These demonstrations are relabelled to generate additional trajectories such that every intermediate state is used as a goal. Next, VaPRL starts collecting data in the training MDP $\mathcal{M}_T$. At every step, VaPRL samples the goal

**Algorithm 1:** Value-Accelerated Persistent Reinforcement Learning (VaPRL)

**Input:** initial state(s) $\mathcal{D}_\rho$, $N$; // N: number of goals for relabeling
**Optional:** Demos $\mathcal{D}$;
Initialize replay buffer $\mathcal{B}, \pi(a \mid s, g), \mathcal{Q}^\pi(s, a, g)$;
// If demos, add them to replay buffer and relabel
$\mathcal{B} \leftarrow \mathcal{B} \cup \mathcal{D}$;
relabel_demos($\mathcal{B}$);
**while** *not done* **do**
    $s \sim \tilde{\rho}$; // sample initial state
    **for** $H_T$ *steps* **do**
        $g \leftarrow G(s, p_g)$ (Eq 5);
        $a \sim \pi(\cdot \mid s, g), s' \sim p(\cdot \mid s, a)$;
        $\mathcal{B} \leftarrow \mathcal{B} \cup \{(s, a, s', g, r(s', g))\}$;
        **for** $i \leftarrow 1, i \leq N$ **do**
            $\tilde{g} \sim \mathcal{D} \cup p_g$; // if $\mathcal{D} = \emptyset$, sample replay buffer
            $\mathcal{B} \leftarrow \mathcal{B} \cup \{(s, a, s', \tilde{g}, r(s', \tilde{g}))\}$;
        update $\pi, Q^\pi$;
        $s \leftarrow s'$;

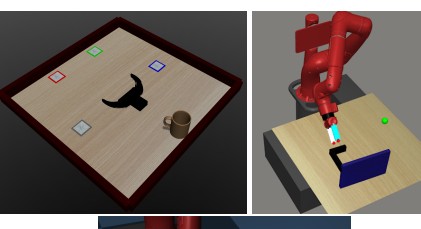

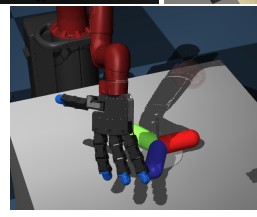

Figure 5: Continuous control environments for goal-conditioned persistent RL. (*top left*) A table-top rearrangement task, where a gripper is tasked with moving the mug to four potential goal positions, (*top right*) a sawyer robot learns how to close the door and (*bottom*) a high-dimensional dexterous hand attached to a sawyer robot is tasked to pick up a three-pronged object.

generator $G$ to get the current goal and collects the next transition using the current policy $\pi$. This transition is added to replay buffer $\mathcal{R}$ along with $N$ relabelled transitions, as described in Sec 4.2. The policy $\pi$ and the critic $Q^\pi$ are updated every step, using any off-policy reinforcement learning algorithm. This loop is repeated for $H_T$ steps till an extrinsic intervention resets the environment to a state $s \sim \tilde{\rho}$. Note, it isn't necessary to initialize the agent close to the goal. Additional details pertaining to the algorithm can be found in the Appendix B.

## 5 Experiments

In this section, we study the performance of VaPRL on continuous control environments for goal-conditioned persistent RL and provide ablations and visualization to isolate the effect of the curriculum. In particular, we aim to answer the following questions:

1. Does VaPRL allow efficient reinforcement learning with minimal episodic resets?
2. How does the scheme for generating a curriculum in VaPRL perform compared to other methods for persistent reinforcement learning?
3. Does VaPRL scale to high dimensional state and action spaces?
4. What does the generated curriculum look like? Is the curriculum effective?

We next describe the specific choices of environments, evaluation metrics and comparisons in order to answer the questions above.

**Environments.** For our experimental evaluation, we consider three continuous control environments, shown in Figure 5. The *table-top rearrangement* is a simplified manipulation environment, where a gripper (modelled as a point mass which can attach to the object if it is close to it) is tasked with taking the mug to one of the 4 potential goal squares. The evaluation horizon is $H_E = 200$ steps and the training horizon is $H_T = 200,000$ steps. This task involves a challenging exploration problem in navigating to objects, picking them up, and dropping them at the right location. The *sawyer door closing* environment involves using a sawyer robot arm to close a door to a particular target angle [45]. For this environment, we set the horizon for evaluation to be $H_E = 400$ and $H_T = 200,000$ steps for training. Since environment resets are not freely available, repeatedly practicing the task implicitly requires the agent to also learn how to open the door. The *hand manipulation* environment, introduced in [14], involves a dexterous hand attached to a sawyer robot. This environment entails a 16 DoF hand that is mounted onto a 6 DoF arm, with the goal of manipulating a 3 pronged object as seen in Figure 5. In particular, the task involves picking up the object from random positions on a table and lifting it to a goal position above the table. This task is particularly challenging since it involves complex contact dynamics with high dimensional state and action spaces. Additionally, the robot has to learn how to reconfigure the object to diverse locations to simulate the test-time conditions where the agent is expected to pickup the object from random locations. For this environment, we set the horizon for evaluation $H_E = 400$ and for training $H_T = 400,000$ steps.

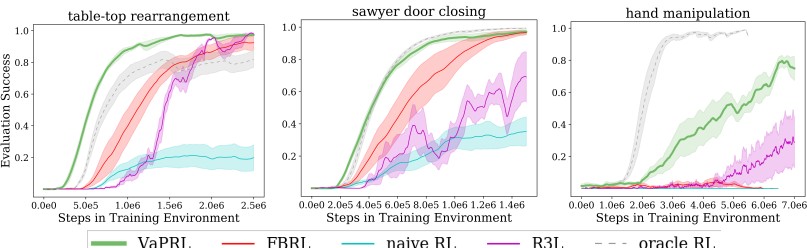

Figure 6: Performance of each method on (*left*) the table-top rearrangement environment, (*center*) the sawyer door closing environment, and (*right*) the hand manipulation environment. Plots show learning curves with mean and standard error over 5 random seeds. VaPRL is more sample-efficient and outperforms prior methods.

**Environment Setup.** For *table-top rearrangement* and *sawyer door closing*, we consider a sparse reward function $r(s, g) = \mathbb{I}(s, g)$, which is 1 when the state $s$ is close to the goal position $g$, and 0 otherwise. Since the *hand manipulation* environment is a substantially more challenging problem, we consider a dense reward function that rewards the the hand and the object to be close to the goal position. To aid exploration in *table-top rearrangement* and *sawyer door closing*, we provide all the algorithms with a small set of trajectories (6 per goal, 3 going from initial state to the goal and the other 3 going in reverse) for each environment, though we do not assume that the trajectories take the optimal path (for example, the trajectories could come from teleoperation in practice). For the *hand manipulation* environment, we provide the agent with 10 trajectories demonstrating the pickup task from random positions on the table and 20 trajectories showing how to reposition the object to different locations on the table. For all environments, we report results by evaluating the number of times the policy successfully reaches the goal out of 10 trials in the evaluation environment $\mathcal{M}_E$ (by resetting to a state from the initial state distribution $\rho$ and sampling an appropriate goal from the goal distribution $p_g$), performing intermittent evaluations as the training progresses. Note, the training agent does not receive the evaluation experience and it is only used to measure the performance on the evaluation environment. Further details about problem setup, demonstrations, implementation, hyperparameters and evaluation metrics can be found in the Appendix.

**Comparisons.** We compare VaPRL to four approaches: (a) A standard off-policy RL algorithm that only trains to reach the goal distribution, such that a new goal $g \sim p_g(s)$ is sampled every $H_E$ steps (labelled **naïve RL**), (b) A forward-backward controller [19, 6] which alternates between $g \sim p_g(s)$ and $g \sim \rho(s)$ for $H_E$ steps each, as described in Section 3 (labelled **FBRL**), (c) A perturbation controller [48] that alternates between optimizing a controller to maximize task reward and a controller to maximally perturb the state via task agnostic exploration (labelled **R3L**), and (d) RL directly on the evaluation environment, resetting to the initial state distribution after every $H_E$ steps (labelled **oracle RL**). This oracle is an expected upper bound on the performance of VaPRL, since it has access to episodic resets. We use soft actor-critic [17] as the base RL algorithm for all methods to ensure fair comparison, although any value-based method would be equally applicable. To emphasize, all the algorithms are provided the same set of demonstrations. Further implementation details can be found in the Appendix.

### 5.1  Persistent RL Results

The performance of each of the algorithms on the three evaluation domains are shown in Figure 6. We see that VaPRL outperforms naïve RL, FBRL and R3L, providing substantial improvements in terms of sample efficiency. For our most challenging domain of hand manipulation, the sample efficiency enables us a reach a much better performance within the training budget. The primary difference between the methods is that VaPRL uses a curriculum of starting states progressing from easier to harder states. In contrast, FBRL always attempts to reach the initial state distribution and R3L uses a perturbation controller to reach novel states in attempt to cover the entire state space uniformly. In the *table-top rearrangement* environment, the agent starts close to the goal and then gradually brings it back to the initial state distribution, trying different intermediate states in the process (discussed in Section 5.3). In the *sawyer door closing* environment, the agent learns to close the door from intermediate angles, incrementally improving the performance. For the hand manipulation domain, the agent focuses on picking up the object from a particular location, and then incrementally grow the locations from which it can complete pickup. In contrast, FBRL chooses attempts the pickup from random states from the initial state distribution $\rho$ and R3L attempts to find new states to pickup the object from (even though it might not be succeeding to pickup the object from previous locations).

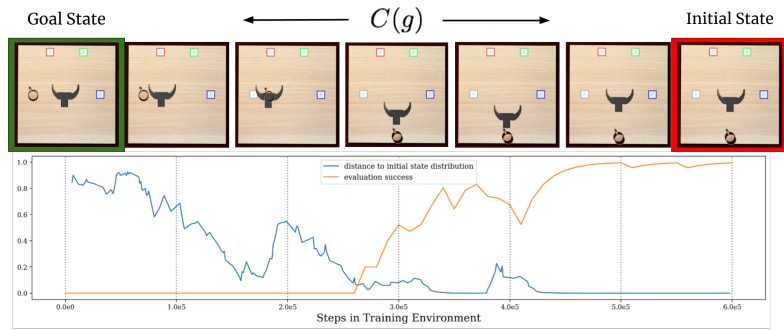

Figure 8: Visualization of curricula generated by VaPRL on the *table-top rearrangement* environment. We plot the *step index distance* between the initial state and the curriculum goals generated by $C(g)$ *(blue)* and the evaluation performance (*orange*) as the training progresses. The distance is normalized to be on the same scale as the success metric, such that a value of 1 corresponds to the test-goal distribution and 0 corresponds to the initial state distribution. We visualize some of the commanded goals $C(g)$ during the training, observing that the curriculum gradually progresses from goal states to initial states with a correlated improvement in evaluation performance.

Compared to oracle RL with resets, VaPRL learns to solve the task while requiring $500\times$ fewer environment resets in the door closing environment, $1000\times$ fewer environment resets in the *table-top rearrangement* environment and dexterous *hand manipulation*. This amounts to less than 20 total interventions for VaPRL, indicating the substantial autonomy with which the algorithm can run.

Surpisingly, in the domains with a sparse reward function (that is, *sawyer door closing* and *table-top reaarrangement*), VaPRL matches or even outperforms the oracle RL method. In the *table-top rearrangement* environment, oracle RL does substantially worse than VaPRL. It has been noted in prior work that multi-goal RL problems can converge suboptimally due to issues arising in the optimization [44]. We hypothesize that an appropriate initial state distribution can ameliorate some of these issues. In particular, moving beyond deterministic initial distributions may lead to better downstream performance (also noted in [47]). For the door opening environment, VaPRL matches the performance of oracle RL. To emphasize, oracle RL is training on the evaluation environment directly, that is $H_T = H_E$ with the environment resetting to a state $s_0 \sim \rho$. In contrast, VaPRL also learns how to reverse the task it is solving and thus only spends half of its training samples collecting the data for the evaluation task (for example, VaPRL learns how to open the door and close it).

## 5.2 Isolating the Role of the Initial State Distribution

We construct an experiment to isolate the effect of the starting state distribution on learning efficiency and downstream performance. In this experiment, the environment resets directly to the state $C(g)$ for VaPRL, such that the policy only has to learn reaching the goals $g \sim p_g$ (labelled **VaPRL + reset**). Analogously, for FBRL, the environment resets to the state $s_0 \sim \rho$, which is identical to the oracle RL method (labelled **oracle RL / FBRL + reset**). For R3L, the environment resets to a state uniformly sampled from the state space (labelled **uniform / R3L + reset**). We run this ablation on the *table-top rearrangement* environment, where the episode horizon for all the algorithms is $H_E = 200$. The results in Figure 7 indicate that the starting state distribution induced by the VaPRL curriculum improves the performance, translating into improved performance in the persistent RL setting.

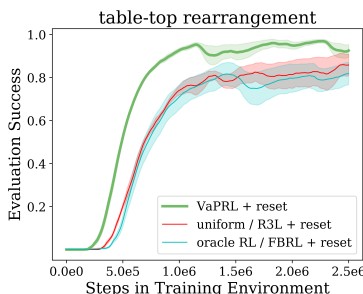

Figure 7: Ablation isolating the effect of curriculum generated by VaPRL.

## 5.3 Visualizations of Generated Curricula

To better understand the curriculum generated by VaPRL, we visualize the sequence of states chosen by Equation 4 as the training progresses on the *table-top rearrangement* environment, shown in Figure 8. As we can observe, initially the curriculum chooses states which are farther away from the initial state and closer to the goal distribution. As training progresses, the curriculum moves towards the initial state distribution. Correspondingly, the evaluation performance starts to improve as we

move closer to the initial state distribution. Thus, VaPRL can generates an adaptive curriculum for the agent to efficiently improve the performance on the evaluation setting.

# 6    Conclusion

In this work, we propose VaPRL, an algorithm that can efficiently solve reinforcement learning problems with minimal episodic resets by generating a curriculum of starting states. VaPRL is able to reduce amount of human intervention required in the learning process by a factor of 1000 compared to episodic RL, while outperforming prior methods. In the process, we also formalize the problem setting of persistent RL to understand current algorithms and aid the development of future ones.

There are a number of interesting avenues for future work that VaPRL does not currently address. A natural extension is to environments with irreversible states. This setting can likely be addressed by leveraging ideas from the literature in safe reinforcement learning [11, 42, 3, 5]. Another extension is to work with visual state spaces, allowing the algorithm to be more broadly applicable in the real world. These two extensions would be a significant step towards enabling autonomous agents in the real world that minimize the reliance on human interventions.

# 7    Disclosure of Funding

This work was supported in part by Schmidt Futures and ONR grant N00014-21-1-2685.

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
