## A   Ergodic MDPs.

As alluded to in Section 3, the formulation discussed in this paper is suitable for reversible environments. For an environment to be considered reversible, we assume that the MDP $\mathcal{M}_E$ is ergodic, as defined in [31]. The MDP is considered ergodic if for all states $a, b \in \mathcal{S}, \exists \pi$, such that $\mathbb{E}_{s \sim \pi(s)|s_0=a}[\mathbb{I}\{s = b\}] > 0$, where $\mathbb{I}$ denotes the indicator function, and $s$ is sampled from the trajectory generated by following policy $\pi$ starting from the state $a$. Any policy which assigns a non-zero probability to all actions will ensure that all states in the environment are visited in the infinite limit for ergodic MDPs, satisfying the condition above.

## B   Implementing VaPRL

VaPRL uses SAC [17] as the base RL algorithm, following the implementation in [18]. Hyperparameters follow the default values: `initial collect steps`: 10,000, `batch size` sampled from replay buffer for updating policy and critic: 256, steps `collected per iteration`: 1, `trained per iteration`: 1, `discount factor`: 0.99, `learning rate`: $3e-4$ (for critics, actors and dual gradient descent used to adjust entropy temperature). The actor and critic network were parameterized as neural networks with two hidden layers each of size 256. The output of the actor network is passed through a `tanh` non-linearity to scale all action dimensions to $[-1, 1]$. Two key differences from the default hyperparameters: The size of the replay buffer was large enough to ensure that none of the collected and relabelled experience is discarded. For `sawyer door closing` and `table-top rearrangement`, the replay buffer has a size of 10M and for `hand manipulation` environment, the replay buffer had a size of 25M. While the weight for entropy is automatically adjusted using dual gradient descent, it was helpful to have a higher initial weight on the reward for environments with a sparse reward function. So, the initial value of temperature $\alpha = 0.1$ for `sawyer door closing` and `table-top rearrangement` (equivalent to reward being scaled 10 times), while for `hand manipulation` environment, the initial value is the default $\alpha = 1$.

VaPRL computes $V^\pi(s) = \mathbb{E}_{a \sim \pi(\cdot|s)}[Q^\pi(s, a)] \approx \frac{1}{L} \sum_{i=1}^{L} Q^\pi(s, a_i)$ where $a_i \sim \pi(\cdot \mid s)$ for $L = 5$. Note, $Q^\pi(s, a) = \mathbb{E}[\sum_{t=0}^{H_E} \gamma^t r(s_t, a_t)]$ needs to be estimated in addition to the critic function estimated by SAC, as the default critic adds the entropy of the policy to the reward while computing the expected sum. $Q^\pi$ (without the compounded entropy) is estimated identically to the default critic otherwise.

As shown in Algorithm 1, there are two instances of goal relabeling: for trajectories collected online and for demonstrations. For every trajectory collected online, VaPRL samples $N$ goals and relabels these trajectories to generate $N$ new trajectories. The goals are sampled from $\mathcal{D} \cup \{g \sim p_g\}$, that is the set of states in the demonstrations and the goal distribution. $N = 4$ is fixed for all environments for online relabeling. A similar scheme to relabel the demonstration set can be followed. However, if the demonstration set is small, a denser of set of relabelled trajectories by using every intermediate state in the trajectory as a goal can be more informative. For a demonstration $\{s_0, s_1, s_2 \ldots s_T\}$, first generate a relabelled trajectory with $s_0$ as the goal, then with $s_1$ and so on to create $T$ new trajectories for every trajectory in the demonstration. VaPRL follows this scheme for `table-top rearrangement` and `sawyer door closing` as these environments only have 6 demonstrations per goal. However, for `hand manipulation environment`, VaPRL reverts to $N = 4$ randomly sampled goals to relabel each demonstration. As there are nearly 30 demonstrations for `hand manipulation` environment, the dense relabeling scheme would produce greater than 1M samples even before any data collection.

VaPRL also uses the demonstrations to compute the distance function $\mathcal{X}_\rho(s)$. For a trajectory going from initial state to the goal $\{s_0, s_1, \ldots s_T\}$, $\mathcal{X}_\rho(s_0) = 0, \mathcal{X}_\rho(s_1) = 1$ and so on. Similarly, if VaPRL has demonstrations going from goal to the initial, it can either exclude them from the curriculum or label them $\mathcal{X}_\rho(s)$ in the reverse order. For `table-top rearrangement` and `sawyer door closing`, VaPRL opts for the latter. For `hand manipulation`, there are no demonstrations corresponding reversing the task, so VaPRL simply excludes the trajectories corresponding to object repositioning from the curriculum. To compute the curriculum goal $C(g)$ in Equation 4, VaPRL minimizes $\mathcal{X}_\rho(s)$ over the set of states where $V^\pi(s, g) \geq \epsilon$. If multiple states minimize $\mathcal{X}_\rho(s)$ while satisfying the constraint, VaPRL chooses a random state amongst the states with minimal $\mathcal{X}_\rho(s)$. If no state satisfies the constraint, VaPRL chooses the state which maximizes $V^\pi(s, g)$.

## C  Experimental Setup

First, we describe the reward functions and the success metrics corresponding to each environment.

`table-top rearrangement`:
$$r(s, g) = \mathbb{I}(\|s - g\|_2 \leq 0.2),$$
where $\mathbb{I}$ denotes the indicator function. The success metric is the same as the reward function. The environment has $4$ possible goal locations for the mug, and goal location for the gripper is in the center.

`sawyer door closing`:
$$r(s, g) = \mathbb{I}(\|s - g\|_2 \leq 0.1),$$
where $\mathbb{I}$ again denotes the indicator function. The success metric is the same as the reward function. The goal for the door and the robot arm is the closed door position.

`hand manipulation`:
$$r(s, g) = 4 \cdot d(h, o) + 10 \cdot d(o, g) + 10 \cdot e^{-d(o,g)^2/0.01} + 10 \cdot e^{-d(o,g)^2/0.001}$$

where $d(h, o) = \|(h_x, h_y, h_z) - (o_x, o_y, o_z)\|_2$ corresponds to the distance between hand and the object and $d(o, g) = \|(o_x, o_y, o_z) - (g_x, g_y, g_z)\|_2$ corresponds to the distance between the object and goal. The reward function encourages the hand to be close to the object and the object to be close to the goal (with higher weight on the object being close to the goal as the coefficient is $10$). The exponential terms are bonuses which are close to $0$ when the object is far from the goal and close to $10$ when the object is close to the goal. The success metric for `hand manipulation` is given by $\mathbb{I}(d(o, g) \leq 0.05)$. The goal for the agent is to bring the object to the center raised $0.2$ meters above the table.

For `sawyer door closing` and `table-top rearrangement`, the environment terminates whenever the agent reaches the goal. Therefore, the maximum return in the environment is $1$. We set the value function threshold $\epsilon = 0.1$ for these environments. For `hand manipulation`, the environment terminates after $H_E = 400$ steps regardless of whether the goal has been achieved. The minimum and maximum return are roughly $-3000$ and $7000$. We set the threshold $\epsilon = -300$.

Next, we discuss the details corresponding to demonstrations for each of the environments.

`table-top rearrangement`: The demonstrations were generated by a human with a discrete action space of {up, down, right, left, grip} which were translated with into noisy continuous actions. We collected $3$ demonstrations taking the mug from the initial state to the goal, and $3$ demonstrations reversing those trajectories, for a total of $24$ demonstrations (as there are $4$ possible goal locations). These demonstrations were sub-optimal as the discrete actions took a smaller step size than the environment allowed (to keep the discrete action space small while still being able to solve the task) and also took sub-optimal route between the initial state and the goals.

`sawyer door closing`: The demonstrations were generated using controller trained via reinforcement learning on an environment with dense rewards and episodic reset interventions. The final demonstrations generated used a noisy version of this learned policy. All methods were provided with $3$ demonstrations closing the door from the initial position and $3$ demonstrations opening the door from the closed position.

`hand manipulation`: The demonstrations for this environment were particularly hard to generate even when using reinforcement learning with episodic reset interventions. We first learned a policy to reposition the object to the center, next we learned a separate policy to raise the object to a height of $0.2$ meters above the center from the center position. We also learned a policy to move the object to different positions on the table. We generated $10$ trajectories from the first two policies, and $20$ trajectories for the last policy and used this as our demonstration dataset. Note, the demonstrations provided are again sub-optimal. In fact, we were not able to provide a single continuous demonstration picking up the object from arbitrary positions on the table (forcing us to train separate policies and collect disjoint demonstrations).

All the baselines use the same hyperparameters and environmental setup as those for VaPRL and have access to same set of demonstrations.

We do not use GPUs for any of the experiments. We used an internal cluster to parallelize the run for different seeds and baselines. The `table-top rearrangement` environment was run for $\sim 12$

hours, the `sawyer door closing` environment was run for 18 hours and `hand manipulation` environment was run for 7 days when using VaPRL and R3L, 5 days when using oracle RL, FBRL and naive RL. We prematurely stopped oracle RL, FBRL and naive RL due to limited computational budget and because the performance of these algorithms showed no signs of further progress (oracle RL had converged while FBRL and naive RL were not improving at solving the task).