# OpenReview forum: "Autonomous Reinforcement Learning via Subgoal Curricula"
_NeurIPS.cc/2021/Conference — NeurIPS 2021 Poster_

### Official Review · Reviewer_2Rka · 2021-07-09

**Rating:** 7
**Confidence:** 4

**Summary:**

This paper focuses on ‘reset-free’ RL, i.e., the aim to eliminate the need for RL agents to be reset to one of a number of states from a start state distribution. The proposed a curriculum inspired approach, where the agent resets to progressively harder starting states, that are further away from the goal. They first formally define the persistent reinforcement learning problem. Then they specify a new method to solve this problem (VaPRL), which indeed outperforms baselines in three simulation experiments.

**Limitations And Societal Impact:**

The authors state there are no explicit negative potential impacts of this work (beyond the general ones of RL), I agree

**Main Review:**

Strong:
* Good formal definition of the type of problem. Although it is close to continual learning, I do agree with the authors that the setting is slightly different, and it is useful to identify this.
* The method is clear and works better than the baselines.

Comments:
* Does the epsilon specification scale with funtion approximation? Then, it may become really hard to trust the epsilon estimate.
* It is intuitively a bit strange that you generate the curriculum by moving start states backwards from the goal, but to find out which state is a good next candidate as start state (using the distance function), you already use V(s,s_0), i.e., the value to get there from the start state. This is kind of the thing you wanted to eliminate with the curriculum in the first place, i.e., only focus on the easier last part first.
* Sec 4.2: I had slight issues understanding what’s going on here. First, how do you know all the elements in the goal space (distribution), are they explicitly given? Second, you start the section about C(g), but then you only relabel the goals right, not the start states? (and why not?)
* Algorithm 1: I would comment again that the switching happens within G(s,g_p), this confused me at first.
* I have some issues with the fact that you give demonstration trajectories to guide initial exploration (L289-305). You wanted to eliminate manual intervention, yet now you start from good trajectories. It is hard for me to judge how optimal these trajectories are, and thereby judge to what extend you are only inpainting/working backwards along an already optimal path with your method.
* There is no discussion at all, which I think is a problem. You should reflect a bit on your method, identify its potential problems and pitfalls, what did you encounter during experimentation.

Conclusion:
This is a decent paper. The problem is relevant, notation and methods are clear, and results are good. My main issue is the use of expert demonstration trajectories to warm-start the algorithm, which I find really hard to judge because I do not know the quality of these demonstration trajectories. They may have a major impact, but the authors do not report on this ablation. A bit more discussion would be helpful al well, but in general this is a good paper with clear contribution, good structure and writing, and good results.

**Time Spent Reviewing:**

1.5

---

> ### Author Response · Authors · 2021-08-10
> **Response**
>
> We are thankful for your detailed review, and judging the paper positively. We address the concerns raised in the review below:
>
> *“My main issue is the use of expert demonstration trajectories … quality of these demonstrations”*
>
> With regards to eliminating manual intervention, we would like to note that collecting demonstrations at the start of training represents a substantially lower cost compared to providing interventions in the form of resets throughout training. Assuming it takes ~1 minute / demonstration, providing 20 demonstrations would take about 20 minutes once before the training. However, robotic learning can take multiple days and providing repeated resets throughout this training would therefore represent a substantially larger cost, as it requires a human to attend to the robot throughout training. Secondly, for several (harder) robotic tasks, demonstrations are sometimes a necessity to learn efficiently and thus using demonstrations does not represent any additional cost in those cases.
>
> To give an estimate of the quality of demonstrations, we run a simple behavior cloning baseline on only the demonstrations and evaluate its average return (averaged over 5 seeds, with standard deviation):
>
> Tabletop rearrangement ->  0. + 0. (max 1., VaPRL ~0.95)
>
> Door closing -> 0.26 + 0.05 (max 1., VaPRL ~0.9)
>
> Hand manipulation -> 0. + 0. (max 1., VaPRL ~0.75)
>
> As can be observed, behavior cloning on the demonstrations performs poorly, confirming that they are not sufficient on their own to solve each task. As is discussed in the paper in Section 4, we can also adapt VaPRL to run without demonstrations. We provide a comparison of VaPRL with and without demonstrations (https://imgur.com/a/s9PFrP4) on the sawyer door closing task. As can be seen, VaPRL still learns to solve the task without demonstrations, albeit slower and slightly worse performance. None of the algorithms could make any learning progress on the other tasks without the provided demonstrations.
>
> To provide more context on their quality, the demonstrations take ~120 steps on an average to solve the task in tabletop-rearrangement while the learned policy takes ~ 50 steps to solve the task. Finally, we would like to emphasize that all baselines used for comparisons were provided with the same set of demonstrations. We agree that the text could benefit from discussion around demonstrations and we will include the experiments and the related discussion in the updated version.
>
> *“Does the epsilon specification scale with function approximation?”*
>
> The choice of epsilon threshold primarily depends on the expected optimal return in the environment. While function approximation may affect the choice of epsilon, we did not investigate this and treated the choice of epsilon as a hyperparameter search. We will add this clarification in the revised paper.
>
> *“It is intuitively a bit strange .... only focus on the easier last part”*
>
> We want to clarify that since we assume access to demonstrations, we use the step index distance function as defined Sec 4.1 L228-L229 to calculate the distance to the initial state distribution. As long as the distance function orders states reasonably, the curriculum would choose the easier starting states at the beginning and progressively increase the difficulty of the task till the initial state distribution.
>
> *“How do you know all the elements in the goal space (distribution), are they explicitly given?”*
>
> Yes, we assume access to a fixed set of samples from the goal distribution apriori to the training. We will update the text to reflect this clarification.
>
> *“Second, you start the section about C(g), but then you only relabel the goals right, not the start states? (and why not?)”*
>
> We assume that the current state $s$, and current goal $g$ to be the input to $\pi(\cdot \mid s, g)$ and $Q^\pi(s, a, g)$ where $a$ is the action. We only consider one unified policy for goals $g$ corresponding to the task *and* goals $g$ for returning to different initial states. For every trajectory collected with a goal $g$, we relabel only the goals with the scheme described in the paper in Sec 4.2, which includes goals corresponding to the task and goals corresponding to different initial states. We do not relabel the initial states themselves as that would misrepresent the underlying dynamics of the environment. Hope this clarification helps!
>
> *"switching happens within G(s,g_p)”*
>
> We will add this comment to Algorithm 1.

---

> > ### Author Response · Authors · 2021-08-26
> > **Additional Comment**
> >
> > We realized that we never responded to your final comment about having more discussion. We will revise the paper to include the following discussion in the conclusion of the paper:
> > VaPRL relies on value function to generate the curriculum for the policy. However, the value function can suffer from severe overestimation in the initial stages of training, generating harder subgoals than what is possible for the policy is capable of solving. Another problem is that persistent exploration can lead the agent afar from the state space that an optimal policy operating in an episodic setting would typically visit. As the state space grows, the agent can drift into more obscure states (consider a robot with several independent objects). This curse of state space dimensionality is a substantially bigger concern in persistent RL as the human intervention to reset the environment is rarely available. Persistent RL with large state spaces would require a different balance of exploration and exploitation (possibly different learning methods too).
> >
> > We will refine this discussion and add more discussion around potential problems and pitfalls in the revised version of our paper.

---

### Official Review · Reviewer_HS6a · 2021-07-12

**Rating:** 5
**Confidence:** 4

**Summary:**

This paper introduces the definition of Persistent Reinforcement Learning (PRL), and an algorithm to solve a goal conditioning version of PRL - Value-Accelerated PRL (VaPRL) - based on two ideas: separate the training and evaluation MDPs; and build a curriculum of (easier to solve) MDPs.

The main idea behind the paper is to solve episodic RL tasks by minimizing the number of resets to the initial state, which is particularly important in real-work RL. This is achieved by using a generator of surrogate goals instead of solving directly for the initial task goal, and by using an initial state distribution different from the evaluation MDP. This initial state distribution is at the beginning of training closer to the goal, and during training it is expanded such that it approaches the initial state distribution of the evaluation MDP.

The experimental results empirically show the claimed benefits, and are compared with other algorithms that do not use any curriculum or have access to resetting to the initial state distribution.

**Limitations And Societal Impact:**

Not applicable.

**Main Review:**

The motivation for this work is very important for real-world reinforcement learning, especially in robotic tasks where resetting to an initial state distribution includes some form of human intervention, either by manually resetting the system, or by engineering a solution to do it automatically.

The paper is well written, organized, easy to follow and read.

Although I like this work, my opinion is that it lacks a more theoretical treatment of the problem and that some claims can be difficult to place in practice in a real system. Nevertheless, I see a good potential for this idea, but I think it is better suited to a robotics conference.

The following are a couple of comments and questions.

Method
----------
A central idea of the paper is to target the sparse reward episodic MDP setting for real scenarios, where the solution of the problem is partly known to humans, e.g., picking up an object. To solve such tasks, the authors claim that placing the agent near the goal is beneficial. While I agree with this statement, there are two concerns. First, for many tasks it is difficult to define what “close to the goal” really means. Second, there are tasks where placing the robot close to the goal is extremely difficult. Take the example of a robot hitting a hockey puck. The reward in this task is sparse and equal to placing the puck in a radius of a given location. One can place the puck near the goal, but if the hockey stick cannot reach it, then it will not be able to solve the task.
“The objective of this goal generator is to recover the optimal policy with respect to J_E efficiently training in an environment with H_T >> H_E”.
If the training environment is different from the evaluation environment, how do you guarantee that the return is maximized in the evaluation environment? From my understanding, the authors define a proxy based on the curriculum training, but unfortunately there are no theoretical guarantees that training on the training MDP will lead to an optimal performance in the evaluation MDP.
By optimizing the policy locally, i.e., from the initial state given by the curriculum to a generated goal, is the trajectory generated by the global policy optimal? Or is it piecewise optimal?
A big focus of the paper is on constructing the curriculum of starting states. Even though it is a good idea, it is in my opinion done in a very heuristic way, and without any guarantees that following this curriculum improves the returns in the evaluation environment.
In the main paper, for the constraint V(s,g) >= epsilon, the authors defined it as belonging to the [0, 1] interval, but in the experiment for hand manipulation, epsilon is out of this range (-300). Could you comment on this?  How to choose epsilon? Is it just a hyperparameter? Can it happen that a “bad” epsilon leads to an underperforming policy?
Goal relabeling is important to reduce the space of possible curriculum goals, and it is solved by sampling N goals at random. My question is again if this choice ensures that we obtain an optimal policy with respect to the evaluation MDP?


Experiments
-----------------
The experiments are well constructed and show the potential of the algorithm in sparse reward environments.
While I understand that NeurIPS is not a robotics conference, it would be good to see how this method applies in reality, especially how easy it is to place the agent near the goal state for a more intricate task.


Citations
------------
The authors seem to be citing a reasonable amount of work from a specific group.
Even though this work is not set in the hierarchical RL framework, it has also some connections to the idea of skill chaining, which although not directly matching this paper it could also be taken into consideration - Konidaris, G. et al, Skill Discovery in Continuous Reinforcement Learning Domains using Skill Chaining, NIPS, 2009

Typos
Eq. (4), the state s is missing under the argmin


**Time Spent Reviewing:**

5

---

> ### Author Response · Authors · 2021-08-10
> **Response**
>
> We are grateful for your thoughtful and detailed review and appreciate the fact that you like the work. We address the concerns raised in the review below:
>
> *“First, for many tasks it is difficult … placing the robot close to the goal is extremely difficult”*
>
> We agree that initializing the system close to the goal is hard in general. However, VaPRL does **not** require this assumption to be in place. In fact, we only initialize the tabletop rearrangement and sawyer door closing environments to be close to the goal. We initialize the most challenging hand manipulation environment at a state from its initial state distribution (the same as the one at the time of evaluation) and **not** close to the goal distribution. We will clarify in Section 4.1 that initializing close to the goal is not a requirement for VaPRL, and we will add the details of environment initialization for every environment in the Appendix. We hope this also addresses the concern related to compatibility of the algorithm with a real system.
>
> *“My opinion is that it lacks a more theoretical treatment”,
> “If the training environment is different .. return is maximized in the evaluation environment?”,
> “without any guarantees that following this curriculum improves the returns in the evaluation environment”*
>
> We would like to note that the paper does not claim to be a theoretical paper. Our primary focus in this work was on providing intuition and empirical validation of the algorithm. However, we do agree that theoretical discussion could be illuminating to the reader and could provide extra validation of the approach. Based on your feedback, we provide some theoretical intuition for why VaPRL would optimize the return in the evaluation MDP. To understand the curriculum and its impact on downstream performance, we want to focus on the initial state distribution generated by it. To simplify this discussion, we first assume an oracle that initializes the agent at the state computed by $C(g)$ and the agent attempts to reach the goal $g$. Under this assumption, the training can be interpreted as a sequence of trajectories starting from the initial state distribution generated by VaPRL. Interestingly, Section 4 in [1] discusses that the optimal policy is preserved even when the initial state distribution is changed in the absence of function approximation. Furthermore, Theorem 6.2 and Section 7.3 in [1] argue for the use of better initial state distributions than the initial state distribution at evaluation, laying the foundation for our work. To understand the properties of the curriculum of initial states more closely, let’s consider some additional assumptions typically made in RL theory: Assume the state and action space to be finite, a tabular setting with no function approximation and assume that we can explore “sufficiently”. For simplicity, further assume that the initial state distribution is $\rho(s) = \delta(s_0)$. Specific to VaPRL, assume that $\epsilon < V^*(s_0)$ and that we have access to an optimal demonstration $s_0, a_0 \ldots s_T = g$ where $a_i \sim \pi^*( \cdot \mid s_i)$ and define $\mathcal{X}_\rho(s_i) = i$. Under these assumptions, we know the Bellman operator is contractive and that value iteration converges to the optimal value function in the limit [2]. Looking at the definition $C(g)$ in equation (4), we know $s_0$ will eventually satisfy the constraint $V^\pi(s, g) \geq \epsilon$, and since $\mathcal{X}_\rho(s_0) = 0$ minimizes the distance function, the curriculum will converge to $s_0$. Once the curriculum converges to $s_0$, the training MDP exactly represents the evaluation MDP and the agent is guaranteed to optimize the return in the evaluation MDP. We hope this simplified discussion provides insight into why VaPRL would lead to the optimal performance in the evaluation MDP. We defer a rigorous analysis with fewer assumptions to future work.
>
> [1] Approximately Optimal Approximate Reinforcement Learning. Kakade & Langford. ICML ‘02.
>
> [2] Dynamic Programming and Optimal Control. Dimitri Bertsekas.
>
> *“In the main paper, for the constraint V(s,g) >= epsilon, the authors defined it as belonging to the [0, 1] interval, but in the experiment for hand manipulation, epsilon is out of this range (-300). Could you comment on this? How to choose epsilon? Is it just a hyperparameter? Can it happen that a “bad” epsilon leads to an underperforming policy?”*
>
> We thank you for noting this discrepancy; the range of epsilon is all real numbers, and we will correct this in the paper. One constraint on epsilon, as noted in the discussion above is that $\epsilon < V^*(s_0)$, which ensures that the states from the initial state distribution can be chosen for resets by the curriculum generator $C(g)$. Choosing a large epsilon can lead to an underperforming policy, as the curriculum would not converge to the initial state distribution. Choosing a small epsilon effectively converts the VaPRL algorithm to the FBRL algorithm. Finally, the hyperparameter search space for epsilon depends on the range of return in the environment (for example, [0, 1] for sparse reward environments). We will add this discussion to the paper.
>
> *“Even though this work is not set in the hierarchical RL framework, it has also some connections to the idea of skill chaining, which although not directly matching this paper it could also be taken into consideration”*
>
> We thank you for bringing up the paper. It is indeed relevant, and we will add a discussion on the suggested paper and skill chaining in general to the paper.
>
> *“I think it is better suited to a robotics conference”*
>
> We would like to kindly note that several papers of similarly empirical nature on this topic have been published in NeurIPS and ICLR. We provide some prior examples:
>
> [1] Continual Learning of Control Primitives: Skill Discovery via Reset-Games. Xu et al. NeurIPS 2020.
>
> [2] Leave no Trace: Learning to reset for Safe and Autonomous Reinforcement Learning. Eysenbach et al. ICLR 2018
>
> [3] The Ingredients of Real-World Robotic Reinforcement Learning. Zhu et al. ICLR 2020.
>
> While we agree that robotics is a well suited application for this work, we believe that there are several algorithmic and theoretical challenges in the process of building systems that can learn autonomously in the real world, which would interest the NeurIPS community.

---

> ### Author Response · Authors · 2021-08-14
> **Any more information?**
>
> Once again, thanks for your detailed and informed review. If your concerns have not been sufficiently addressed, is there any additional information or clarification we can provide to do so?

---

### Official Review · Reviewer_Ta14 · 2021-07-16

**Rating:** 7
**Confidence:** 4

**Summary:**

The paper proposes a curriculum approach for persistent goal conditioned reinforcement learning, that is, for learning how to train continuously and then evaluate in episodic fashion. The main idea is, in each iteration, to try to keep the value of moving from a selected state to the goal above a threshold while selecting the state (where the agent tries to move before moving to the goal) as close as possible to the initial state distribution. The proposed approach outperforms comparison methods in three different simulated robotic benchmark tasks.

**Limitations And Societal Impact:**

Yes

**Main Review:**

According to my knowledge persistent learning is a not so well researched problem setting and thus the proposed approach is novel. The robotics motivation is definitely valid. Related work appears to be adequately cited.

The submission appears technically sound. The experimental results show that in three different simulated robotic goal-conditioned tasks the proposed algorithm outperforms baselines.

The proposed algorithm is based on generating a state that is as close as possible to the initial state distribution while maintaining predicted value over a threshold. The goal generator then uses that state as the current goal until reaching that state and going towards the goal of the actual task. Distance is measured using the number of time steps from the state to the initial state from existing data.

An interesting question is how the discount factor influences the behavior of the algorithm. Compared to continuous learning, since the agent always goes to the same goal and the same task repeats, here long term planning and thus the discount factor are not as important?

The paper makes the assumption of a reversible environment but this is adequately discussed.

Computation time should be reported in the paper since the approach does computations that the comparison methods do not.

The paper is well written and organized. There are some typos that should be corrected. Examples:

Line 79: "a task-graph uses the current state to decides" -> "a task-graph uses the current state to decide"
Line 292: "the the" -> "the"
Line 370: "Thus, VaPRL can generates" -> "Thus, VaPRL generates"

The results are important for real robotic reinforcement learning. Training without resets is critical in many robotic tasks. The proposed approach appears efficient which is validated in three benchmark tasks.

Update:

I was satisfied with the rebuttal and will keep my score.



**Time Spent Reviewing:**

5

---

> ### Author Response · Authors · 2021-08-10
> **Response**
>
> We are grateful for your review and are happy that you are positively disposed to the paper. We address the concerns raised in the review below:
>
> *“Computation time should be reported in the paper since the approach does computations that the comparison methods do not.”*
>
> We provide the computation times for different approaches, executed on the same CPU-only machine:
>
> FBRL: 0.0063 secs/step
>
> VaPRL: 0.0071 secs/step
>
> R3L: 0.0066 secs/step
>
> The numbers here reflect the time per step collected in the environment in simulation (which includes all the computation and simulating the next step). However, we note that real-world training is usually bottlenecked by data collection time which dwarfs the computation time. Moreover, in practical systems, the computations can be parallelized with data collection. Therefore, the additional computation cost of VaPRL likely does not translate into very significant practical downsides.
>
> We appreciate you pointing out the typos; we will fix them in the next revision and double check the rest of the paper to fix any other remaining typos.

---

### Official Review · Reviewer_yCgQ · 2021-07-16

**Rating:** 6
**Confidence:** 3

**Summary:**

The authors present a novel RL method that learns an adaptive curriculum to minimize the number of resets needed in episodic RL. This is evaluated against competing algorithms on manipulation tasks.

**Limitations And Societal Impact:**

Superficial but adequate.

**Main Review:**

Minimizing the number of resets needed could be useful in many real-world RL applications. The approach is interesting and the experimental results look very promising compared to competing approaches.

My main concern is with the presentation. The paper is mostly well-written, but there appears to be many complicated design choices in there with subtle theoretical implications. I'm only superficially familiar with the related work in this niche of RL, but the general idea and mathematical formalization could be clearer. I had to read it several times.

Some comments:
- If the focus is *reducing* resets (see main contribution on L54), it was a bit confusing why the main baseline wasn't the number of resets needed vs. episodic RL? Instead you compare against "episodic" RL without resets, which is also a bit counterintuitive.
- It is not obvious to me why RL without resets performs so poorly in Fig.2. You claim "This is because these methods rely on the ability to sample the initial state distribution arbitrarily", but this does not seem entirely true as RL can be non-episodic (and many "episodic" applications just use approximate infinite horizons via e.g. GAE). Could you comment on why this happens? Is it perhaps because your learned reversion to the initial state distribution yields better exploration somehow (e.g. if items otherwise get pushed out to the corners with time), or could it just be that you are using HER?
- Likewise in the formal problem definition in (3), it's a bit confusing to use a finite horizon formulation when considering an infinite horizon setting in Fig.2 and calling it "persistent" RL. It really seems to me that this is an inifinite horizon problem. It's not that you are minimizing resets, you want to maximize reward over an infinite horizon without resets at all, right? It would help to at least clarify this.
- The abstract and intro also seems to equate all RL with episodic RL. Again, careful with terminology and notation.

Minor details:

L66: "can be up to 30% more sample-efficient" - needlessly vague, were on average x% more sample-efficient in our experiments perhaps?

L70: manually design*ed*

L246 & 252: Conenct reference [2] to HER.

L252: "... and *the* goal ..."

L292: "the the"

----


After rebuttal:
I thank the authors for their clear response. I share some of the other reviewers' concerns that this paper could benefit from a more formal treatment, but the authors' clarifications were helpful. I will maintain my score of 6.

**Time Spent Reviewing:**

5

---

> ### Author Response · Authors · 2021-08-10
> **Response**
>
> We are thankful for your comments. We address your concerns below:
>
> *“it's a bit confusing to use a finite horizon formulation when considering an infinite horizon setting in Fig.2 and calling it "persistent" RL. It really seems to me that this is an infinite horizon problem. It's not that you are minimizing resets, you want to maximize reward over an infinite horizon without resets at all, right? It would help to at least clarify this”*
>
> This interpretation is partly correct. You are correct that we are not explicitly minimizing resets, but our goal isn’t to maximize the reward over an infinite horizon, which would correspond to a lifelong reinforcement learning setting. Our goal instead is to use a MDP with low-frequency resets (potentially none in the infinite horizon case) to acquire the ability to execute a specified task from a particular initial state distribution. As an example, consider the problem of a robot learning how to close a door. We set up a training environment where the robot can repeatedly practice and improve its ability to close the door. We do not measure the robot’s ability to close the door in its training MDP, as repeatedly practicing how to close the door requires the robot to try several things, including opening the door to set up the next iteration of practice. We measure the ability of the robot to close the door in its evaluation MDP, which corresponds to the performance when the policy will be deployed after training. In some sense, this represents a transfer learning problem where the training algorithm has access to a training MDP to learn how to do well on the evaluation MDP. We consider this transfer learning problem to be persistent RL, which is different from maximizing the return over an infinite horizon. There is some pertinent discussion in Section 2 L[103-113], but we will also include this clarification in the revision.
>
> *“If the focus is reducing resets (see main contribution on L54), it was a bit confusing why the main baseline wasn't the number of resets needed vs. episodic RL?”*
>
> The primary goal of this work is to learn efficiently under realistic training assumptions, i.e. when human interventions for resetting environments are sparsely available. Therefore, we set up persistent RL to evaluate the efficiency and performance of the algorithms when resets are sparsely available, that is when $H_T \gg H_E$. Nonetheless, we have added a comparison to FBRL with varying numbers of resets (i.e. with varying $H_T$) in the tabletop-rearrangement environment. We provide the number of samples and resets required to reach an average performance of 0.5 (=50% success) as the training horizon $H_T$ is varied (full learning curves available here: https://imgur.com/a/mVE1zHt):
>
> VaPRL -> 550k samples, 3 resets ($H_T$ = 200k)
>
> FBRL -> 730k samples, 3650 resets ($H_T$ = 200)
>
> FBRL ->  930k samples, 2000 resets ($H_T$ = 2k)
>
> FBRL -> 1.03M samples, 52 resets ($H_T$ = 20k)
>
> FBRL -> 1.1M samples, 6 resets ($H_T$ = 200k)
>
> As can be seen, decreasing the number of resets slows the performance, and VaPRL provides the same performance much faster with much fewer resets.
>
> *“It is not obvious to me why RL without resets performs so poorly in Fig.2”*
>
> We agree that this is not an obvious finding, and we will update the paper to provide the following intuition about the door closing example task. Naively maximizing reward in a training MDP without resets leads the agent to simply leave the door closed once it reaches that position, even if it has not collected enough training experience needed to reliably close the door from the initial state distribution. As a result, when it is evaluated on the skill of closing the door, it cannot effectively solve the task, and consequently receives low reward in the evaluation MDP, as illustrated on the y-axis of Fig.2. This phenomenon was also noted in [1].
>
> [1] The Ingredients of Real-World Robotic Reinforcement Learning. Zhu et al. ICLR 2020.
>
> We’ll also revise the abstract and the introduction to clarify that the expressed limitations pertain to episodic RL, and we will also correct the minor changes suggested in the revised paper.

---

### Decision · Program_Chairs · 2021-09-27

**Decision:**

Accept (Poster)

**Comment:**

After reading each other's reviews and the authors' feedback, the reviewers discussed the merits and flaws of the paper.
The reviewers did not reach a consensus. In particular, the lack of a theoretical analysis together with some doubts about its applicability to realistic scenarios have been considered as significant limitations.
Nonetheless, the majority of the reviewers were satisfied with the answers provided by the authors and think that the approach proposed in the paper is interesting and promising.
Overall, I think that, despite its limits, the paper provides a nice contribution and I propose to accept it.
I want to congratulate the authors and invite them to modify their paper following the reviewers' suggestions.